# Decomposing 3D Scenes into Objects via Unsupervised Volume Segmentation

## Abstract

We present ObSuRF, a method which turns a single image of a scene into a 3D model represented as a set of Neural Radiance Fields (NeRFs), with each NeRF corresponding to a different object. A single forward pass of an encoder network outputs a set of latent vectors describing the objects in the scene. These vectors are used independently to condition NeRF decoders, defining the geometry and appearance of each object. We make learning more computationally efficient by deriving a novel loss, which allows training NeRFs on RGB-D inputs without explicit ray marching. After confirming that the model performs equal or better than state of the art on three 2D image segmentation benchmarks, we apply it to two multi-object 3D datasets: A multiview version of CLEVR, and a novel dataset in which scenes are populated by ShapeNet models. We find that after training ObSuRF on RGB-D views of training scenes, it is capable of not only recovering the 3D geometry of a scene depicted in a single input image, but also to segment it into objects, despite receiving no supervision in that regard.

## 1 Introduction

The ability to recognize and reason about 3D geometry is key to a wide variety of important robotics and AI tasks, such as dynamics modelling, rapid physical inference, robot grasping, or autonomous driving (Battaglia et al., 2013; Chen et al., 2016; Mahler et al., 2019; Driess et al., 2020; Li et al., 2020). While progress has been made on integrating traditional representations of geometry with machine learning systems (Wang et al., 2018; Qi et al., 2017; Nash et al., 2020), they remain difficult to work with due to either undesirable scaling behavior (voxels, point clouds) or discrete structures and an inability to train on observational data (polygonal meshes).

Recent work has revealed coordinate-based neural networks as an alternative. These representations are continuous, independent of resolution or scene dimensions, and can directly map 3D coordinates to binary occupancy indicators (Mescheder et al., 2019), signed distances to the shape (Park et al., 2019), or volume radiances (Mildenhall et al., 2020). When used with an encoder network, such functions can facilitate learning low-dimensional *implicit* representations of geometry (Mescheder et al., 2019; Kosiorek et al., 2021). They can also render high quality images (Martin-Brualla et al., 2021; Barron et al., 2021), and synthesize novel scenes (Guo et al., 2020; Niemeyer & Geiger, 2020).

In this paper, we investigate a different aspect. We ask whether a latent-variable model built around implicit representations may be used to *infer* semantically meaningful information from a given image, without human supervision. Such representations may then be used for downstream reasoning tasks. In particular, we focus on obtaining object-based representations from multi-object scenes in an unsupervised way. Representations factored into objects are beneficial for dynamics modelling, visual question answering, and many other tasks (Battaglia et al., 2016; Santoro et al., 2017; Battaglia et al., 2018). Previous works in unsupervised object-based representation learning have mostly focused on segmenting 2D images (Eslami et al., 2016; Greff et al., 2020; Locatello et al., 2020). Such methods have remained limited to visually simple, mostly synthetic data, whereas more realistic scenes with complex textures and geometries have remained out of reach (Weis et al., 2020).

To move towards more complex scenarios, we present *ObSuRF*, a model which learns to decompose scenes consisting of multiple *Objects* into a *Superposition of Radiance Fields*. We first encode the input image with a slot-based encoder similar to (Locatello et al., 2020), but use the resulting set of latent codes to condition continuous 3D scene functions (Mildenhall et al., 2020; Kosiorek et al.,

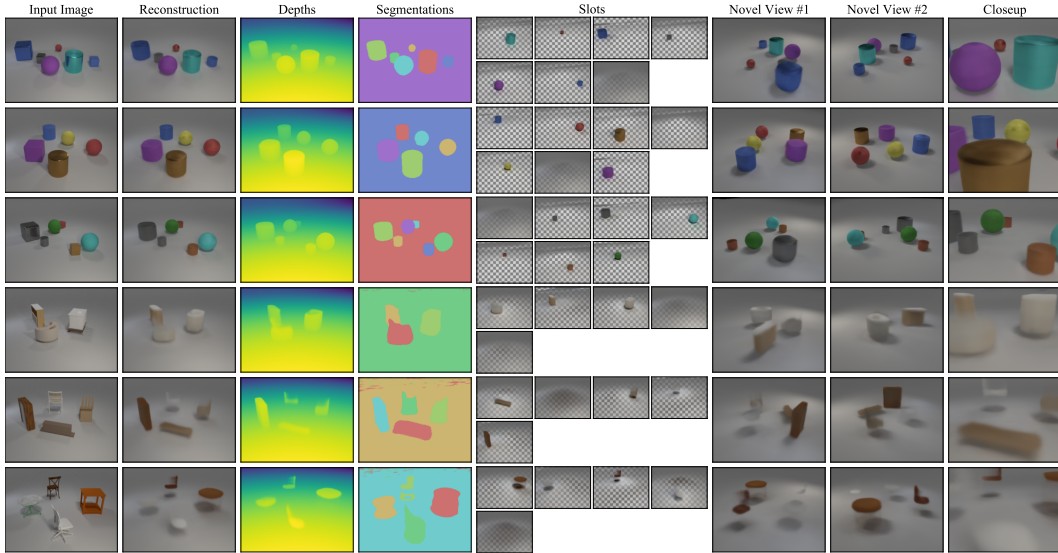

Figure 1: ObSuRF uses a single image (left) to infer a set of NeRFs (slots) representing different objects. ObSuRF accurately models the geometry and appearance of CLEVR-3D scenes (top), and decomposes them into objects. It also produces details which are not visible in the input image, such as the full shape of partially occluded objects, or the shadows on the backsides of objects. This allows rendering the scene from arbitrary angles (right). The MultiShapeNet dataset (bottom) features a much larger variety of more complicated objects, such that ObSuRF cannot capture all of its details. However, it still learns to segment the scene into volumes corresponding to the objects, and to reproduce their dimensions, color, and position. For a full demo, please watch the supplementary videos at https://sites.google.com/view/obsurf/.

2021) instead of directly generating 2D images. For training the model in 3D, we provide three RGB-D views of each scene, and optimize the model to match the observed depths and colors. To do so, we reframe NeRF's volumetric rendering as a Poisson process and derive a novel training objective, which allows for more efficient training when depth information is available as supervision. After confirming that the resulting model is capable of segmenting 2D images as well or better as previous approaches, we test it on two new 3D benchmarks: A 3D version of CLEVR (Johnson et al., 2017) featuring multiple viewpoints, camera positions, and depth information, and MultiShapeNet, a novel multiobject dataset in which the objects are shapes from the ShapeNet dataset (Chang et al., 2015). We will publish both of these benchmarks and the code for our model with this paper.

## 2 RELATED WORK

**Learned Implicit Representations of 3D Shapes.** Early work focused on predicting the geometry of shapes using occupancy fields (Mescheder et al., 2019) or signed distance functions (SDFs) (Park et al., 2019). These models were trained using ground-truth geometry, for instance by sampling points from the meshes in the ShapeNet dataset (Chang et al., 2015). Mildenhall et al. (2020) subsequently introduced Neural Radiance Fields (NeRF), which predict density and color of points in a scene. This admits rendering via differentiable ray tracing through the scene. The required supervision is thereby reduced to a small set of posed images. However, ray tracing makes this approach very computationally expensive. Therefore, in this work we opt for a pragmatic middle ground: we assume access to views of the training scenes enriched with depth information (RGB-D), and show how this allows for training NeRFs without expensive raymarching. We note that unlike ground-truth meshes, such data may be obtained at scale in the real world with reasonable accuracy using either low-cost camera systems (Horaud et al., 2016) or by estimating depth from stereo vision or motion, see e.g. (Chang & Chen, 2018; Teed & Deng, 2020).

In its original form, NeRF is fitted to a single scene, which makes it too slow for real-time applications, and also ill-suited for representation learning, as the scene is encoded in neural network

weights. Recent work has used encoded representations of input scenes to condition a decoder shared between scenes (Yu et al., 2021a; Trevithick & Yang, 2020; Kosiorek et al., 2021). We take a similar approach, but put a greater emphasis on compact and factored representations. Using depth supervision to improve NeRF training has concurrently been proposed by Deng et al. (2021), without however avoiding ray marching and the corresponding performance costs.

**Unsupervised Image Segmentation.** The work in this area can be broadly grouped in two categories. Patch-based approaches represent scenes as a collection of object bounding boxes (Eslami et al., 2016; Kosiorek et al., 2018; Stelzner et al., 2019; Crawford & Pineau, 2019). This encourages spatial consistency in objects, but limits the model's ability to explain complicated objects. Scene-mixture models represent scenes as pixelwise mixture models, with each mixture component corresponding to a single object (Greff et al., 2016; 2017; van Steenkiste et al., 2018; Burgess et al., 2019; Greff et al., 2020; Locatello et al., 2020; Engelcke et al., 2020; 2021). Since spatial consistency is not enforced, these models sometimes segment images by color rather than by object (Weis et al., 2020), motivating combinations of the two concepts (Lin et al., 2020). We mitigate these issues by employing NeRF. Explicitly formulated as an MLP of Fourier-encoded spatial coordinates (instead of a CNN, similarly to e.g. Kabra et al. (2021)), it is naturally biased toward spatial coherence. We note that no model in this line of research has been shown to work well on general natural images. Instead they have targeted synthetic benchmarks or constrained real world environments. Some recent models use viewpoint-dependent decoders (Nanbo et al., 2020; Chen et al., 2021; Kabra et al., 2021), which allows generating images from novel viewpoints. Since there is nothing to enforce multi-view consistency, these models are unlikely to generalize to novel viewpoints.

**Unsupervised 3D Segmentation.** Most work in this area focuses on segmenting single shapes into parts, and is trained on ground truth geometry. BAE-NET (Chen et al., 2019) reconstructs voxel inputs as a union of occupancy functions. CvxNet (Deng et al., 2020) and BSP-Net (Chen et al., 2020) represent shapes as a union of convex polytopes, with the latter capable of segmenting them into meaningful parts. Similarly, UCSG-Net (Kania et al., 2020) combines SDF primitives via boolean logic operations, yielding rigid but interpretable shape descriptions.

Only a small number of publications attempt to segment full scenes into objects. Ost et al. (2021) train a model to segment a complex real-world video into objects, but doing so requires a full scene graph, including manually annotated tracking information such as object positions. In GIRAFFE (Niemeyer & Geiger, 2020) object-centric representations emerge when a set of NeRFs conditioned on latent codes is trained as a GAN. However, the focus lies on image *synthesis* and not *inference*. Consequently, it is not straightforward to obtain representations for a given scene. BlockGAN (Nguyen-Phuoc et al., 2020) achieves similar results, but uses latent-space perspective projections and CNNs instead of NeRFs. RELATE (Ehrhardt et al., 2020) builds on this to allow the manipulation of scenes. Elich et al. (2021) present a model which encodes a single input image into a set of deep SDFs representing the objects in the scene. In contrast to our work, theirs requires pretraining on ground-truth shapes and operates on simpler scenes without shadows or reflections. Closest to our work is a system concurrently developed by Yu et al. (2021b), which also learns to segment scenes into a set of NeRFs using a slot based encoder. However, they train using NeRF rendering and mitigate its computational cost by adjusting the rendering resolution.

**Set Prediction.** Objects present in a scene are interchangeable and form a set. Any attempt to assign a fixed ordering to the objects (such as left-to-right) will introduce discontinuities as they switch positions (e.g. as they pass behind each other) (Zhang et al., 2020). It is therefore essential to use *permutation equivariant* prediction methods in such settings. We build on recent works which construct permutation equivariant set prediction architectures (Locatello et al., 2020; Goyal et al., 2021; Kosiorek et al., 2020; Stelzner et al., 2020).

# 3 METHODS

ObSuRF decomposes 3D scenes into multiple NeRFs, each representing a separate object. We derive a principled way of composing multiple NeRFs into a single, albeit more expensive (by a factor of $n$), scene function. To decrease this cost, we show how to make NeRF integration cheaper when ground-truth depth is available at training time.

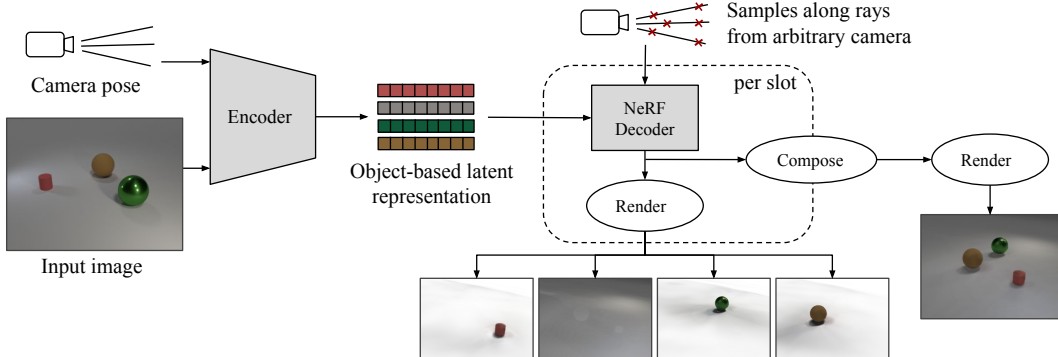

Figure 2: ObSuRF architecture. The encoder is given an input image with the corresponding camera pose, and infers an object-based latent representation consisting of slots. These slots are used independently to condition a shared NeRF decoders. The volumes represented by the resulting scene functions may be rendered individually by querying them along rays coming from an arbitrary camera. Alternatively, the volumes may be composed in order to render the full scene.

### 3.1 NEURAL RADIANCE FIELDS (NERFS)

NeRFs (Mildenhall et al., 2020) represent the geometry and appearance of a scene as a neural network $f : (\mathbf{x}, \mathbf{d}) \to (\mathbf{c}, \sigma)$ mapping world coordinates $\mathbf{x}$ and viewing direction $\mathbf{d}$ to a color value $\mathbf{c}$ and a density value $\sigma$. To guarantee coherent geometry, the architecture is chosen such that the density $\sigma$ is independent of the viewing direction $\mathbf{d}$. To simplify notation, we refer to the individual outputs of $f(\cdot)$ as $\sigma(\mathbf{x})$ and $\mathbf{c}(\mathbf{x}, \mathbf{d})$. NeRFs allow rendering images using classic ray marching techniques for volume rendering (Blinn, 1982; Kajiya & von Herzen, 1984). Specifically, the color $\hat{C}(\mathbf{r})$ corresponding to the ray $\mathbf{r}$ is given by

$$\hat{C}(\mathbf{r}) = \int_0^\infty T(t)\sigma(\mathbf{r}(t))c(\mathbf{r}(t), \mathbf{d})dt\,, \tag{1}$$

with transmittance $T(t) = \exp\left(-\int_0^t \sigma(\mathbf{r}(t'))dt'\right)$. Since this procedure is differentiable, NeRFs are trained by minimizing the L2 loss between the rendered colors $\hat{C}(\mathbf{r})$ and the colors of the training image $C(\mathbf{r})$, i.e. $\mathcal{L}_{\text{NeRF}}(\mathbf{r}) = \|\hat{C}(\mathbf{r}) - C(\mathbf{r})\|_2^2$. This is expensive, however, as approximating Eq. (1) requires evaluating $f$ many times per pixel (256 in Mildenhall et al. (2020)). We will show that, if depth is available, we can instead train NeRFs with just two evaluations per pixel, reducing the required computation by a factor of 128 per iteration[1].

### 3.2 VOLUME RENDERING AS A POISSON PROCESS

While ray marching in Eq. (1) allows computing the ray color, we are also interested in finding the distribution of distances that give rise to this color—this will allow using depths in RGB-D data for supervision. To this end, we show that Eq. (1) derives from an *inhomogenous spatial Poisson process* (Blinn, 1982; Møller & Waagepetersen, 2003).

Consider a ray $\mathbf{r}(t) = \mathbf{x_0} + \mathbf{d}t$ traveling through a camera at position $\mathbf{x_0}$ along direction $\mathbf{d}$. The probability that light originating at point $\mathbf{r}(t)$ will not scatter and reach the camera unimpeded (transmittance) is equal to the probability that no events occur in the spatial Poisson process $T(t) = \exp\left(-\int_0^t \sigma(\mathbf{r}(t'))dt'\right)$. NeRF does not model light sources explicitly, but assumes that lighting conditions are expressed by the color value of each point. As a result, the light emitted at the point $\mathbf{r}(t)$ in the direction of the camera $-\mathbf{d}$ has the color $\mathbf{c}(\mathbf{r}(t), \mathbf{d})$, and its intensity is proportional to the density of particles $\sigma(\mathbf{r}(t))$ at that point. Consequently, the amount of light *reaching* the camera from $\mathbf{r}(t)$ is proportional to

$$\mathrm{p}(t) = \sigma(\mathbf{r}(t))T(t)\,. \tag{2}$$

---

[1]This reduction affects only the decoder side of out autoencoder model.

In fact, under mild assumptions, $p(t)$ is exactly equal to the distribution of possible depths $t$ at which the observed colors originated[2]. We provide the full derivation in Appendix B.1.

We can now reframe Eq. (1) as the expected color value under the depth distribution $p(t)$:

$$\hat{C}(\mathbf{r}) = \mathbb{E}_{t\sim p(\cdot)}[\mathbf{c}(\mathbf{r}(t), \mathbf{d})] = \int_0^\infty p(t)c(\mathbf{r}(t), \mathbf{d})dt\,. \tag{3}$$

Typically, one chooses a maximum render distance $t_{\text{far}}$ as the upper bound for the integration interval. This leaves the probability $p(t > t_{\text{far}}) = T(t_{\text{far}})$ that light from beyond $t_{\text{far}}$ is missed. To account for that, Eq. (3) is renormalized by dividing by $T(t_{\text{far}})$ after approximating it with samples $0 \le t_i \le t_{\text{far}}$.

### 3.3 FAST TRAINING ON DEPTH AND COLORS

We now turn to the optimization objectives we use to train ObSuRF. Even though Mildenhall et al. (2020) use a hierarchical sampling scheme, training NeRFs requires many function evaluations for each ray and is therefore extremely expensive. Moreover, as we show in Appendix B.2, this sampling produces a biased estimate for $\hat{C}(\mathbf{r})$, and, consequently $\mathcal{L}_{\text{NeRF}}$. This is caused by the nested integration in Eq. (1): If too few samples are collected, there is a significant chance that thin, high-density volumes are missed entirely, even though they would dominate the color term $\hat{C}(\mathbf{r})$ if it were to be evaluated analytically (see e.g. Rainforth et al. (2018) for a thorough treatment of such issues).

To avoid this computational cost during training, we use depth supervision by training on RGB-D data, i.e. images for which the distance $t$ between the camera and the visible surfaces is known. Instead of integrating over $t$, this allows us to directly maximize the depth log-likelihood $\log p(t)$ in Eq. (2) for the known values of $t$. While we still need to approximate the inner integral, an unbiased estimate can be obtained with uniform random samples from $q(\cdot) = \text{Uniform}(0, t_{\text{far}})$ as $\log p(t) = \log \sigma(\mathbf{r}(t)) - t\mathbb{E}_{t'\sim q(\cdot)}[\sigma(\mathbf{r}(t'))]\,.$ Since $\sigma(\mathbf{r}(t'))$ is likely to be large near the surface $t$ and close to zero for $t' \ll t$, the variance of this estimator can be very large—especially at the beginning of training. To reduce the variance, we importance sample $t'$ from a proposal $q'(\cdot)$ with a higher density near the end of the integration range,

$$\log p(t) = \log \sigma(\mathbf{r}(t)) - \mathbb{E}_{t'\sim q'(\cdot)}[\sigma(\mathbf{r}(t'))/q'(t')]\,. \tag{4}$$

In practice, we set $q'(\cdot)$ to be an even mixture of the uniform distributions from $0$ to $0.98t$ and from $0.98t$ to $t$, i.e. , we take $50\%$ of the samples from the last $2\%$ of the ray's length.

We fit the geometry of the scene by maximizing the depth log-likelihood of Eq. (4). Staying with our probabilistic view, we frame color fitting as maximizing the log-likelihood under a Gaussian distribution. Namely, since we know the point $\mathbf{r}(t)$ at which the incoming light originated, we can evaluate the color likelihood as $p(C \mid \mathbf{r}(t), \mathbf{d}) = \mathcal{N}(C \mid \mathbf{c}(\mathbf{r}(t), \mathbf{d}), \sigma_C^2)$ with fixed standard deviation $\sigma_C$. Overall, evaluating the joint log-likelihood $\log p(t, C)$ requires only two NeRF evaluations: At the surface $\mathbf{r}(t)$ and at a point $\mathbf{r}(t')$ between the camera and the surface. In practice we take the surface sample at $\mathbf{r}(t+\epsilon)$ with $\epsilon \sim \text{Uniform}(0, \delta)$. This encourages the model to learn volumes with at least depth $\delta$ instead of extremely thin, hard to render surfaces.

### 3.4 COMPOSING NERFS

We are interested in segmenting scenes into objects by representing each of them as an independent NeRF. A scene is then represented by a set of NeRFs $f_1, \ldots, f_n$, each with densities $\sigma_i$ and colors $\mathbf{c}_i$. We now show how to compose these NeRFs into a single scene function. We arrive at the same equations as (Niemeyer & Geiger, 2020; Guo et al., 2020; Ost et al., 2021), but derive them by treating the composition of NeRFs as the *superposition* (Møller & Waagepetersen, 2003) of independent Poisson point processes, yielding a probabilistic interpretation of the algorithm.

Specifically, we assume that, while we move along the ray $\mathbf{r}$, each of the $n$ Poisson processes has a chance of triggering an event independently. The total accumulated transmittance $T(t)$ from $\mathbf{r}(t)$

---

[2]Note that the transmittance is symmetric, and that both occlusion and radiance depend on the density $\sigma(\cdot)$. Therefore, if a light source is present at $\mathbf{x_0}$, the distribution of points along $\mathbf{r}$ it illuminates is also given by Eq. (2). It can be convenient to think of ray marching as shooting rays of light from the camera into the scene instead of the other way around.

to $\mathbf{x_0}$—the probability of encountering no events along the way—should therefore be equal to the product of the transmittances $T_i(t)$ of each process:

$$T(t) = \prod_{i=1}^{n} T_i(t) = \exp\left(-\int_0^t \sum_{i=1}^{n} \sigma_i(\mathbf{r}(t))dt'\right). \quad (5)$$

This is equivalent to another Poisson process with density $\sigma(\mathbf{x}) = \sum_i \sigma_i(\mathbf{x})$.

To compute the color value, we additionally need to determine to what degree each of the component NeRFs is responsible for the incoming light. Following Eq. (2), the probability that NeRF $i$ is responsible for the light reaching the camera from depth $t$ is $\mathrm{p}(t, i) = \sigma_i(\mathbf{r}(t))T(t)$. Similarly to Eq. (3), we compute the pixel color by marginalizing over both depth $t$ and component $i$, yielding

$$\hat{C}(\mathbf{r}) = \mathbb{E}_{t,i\sim\mathrm{p}(\cdot)}[\mathbf{c}_i(\mathbf{r}(t), \mathbf{d})] = \int_0^\infty \sum_{i=1}^{n} \mathrm{p}(t, i)c_i(\mathbf{r}(t), \mathbf{d})dt. \quad (6)$$

It can also be useful to marginalize only one of the two variables. Marginalizing $i$ yields the depth distribution $\mathrm{p}(t) = \sum_i \mathrm{p}(t, i)$ which we use to render scenes via hierarchical sampling as in (Mildenhall et al., 2020). By marginalizing $t$, one obtains a categorical distribution over the components $\mathrm{p}(i) = \int_t \mathrm{p}(t, i)$, which we use to draw segmentation masks. Finally, in order to compute the color loss derived above, we use the expected color $\mathbf{c}(\mathbf{x}, \mathbf{d}) = \sum_i \mathbf{c}_i(\mathbf{x}, \mathbf{d})\sigma_i(\mathbf{x})/\sigma(\mathbf{x})$.

## 3.5 LEARNING TO ENCODE SCENES AS SETS OF NERFS

We now describe the full encoder-decoder architecture of ObSuRF (see Figure 2), which can *infer* scene representations from a single view of that scene—for *any* scene from a large dataset. ObSuRF then turns that representation into a set of NeRFs, which allows rendering views from arbitrary viewpoints and provides full volumetric segmentation of the scene.

The encoder network $f_{\mathrm{enc}}$ infers a set of latent codes $z_1, \ldots, z_n$ (called slots) from the input image and the associated camera pose. Each slot represents a separate object, the background, or is left empty. By using a slot $z_i$ to condition an instance of the decoder network $f_{\mathrm{dec}}$, we obtain the component NeRF $f_i(\cdot, \cdot) = f_{\mathrm{dec}}(\cdot, \cdot; z_i)$. In practice, we set the number of slots $n$ to one plus the maximum number of objects per scene in a given dataset to account for the background.

**Encoder.** Our encoder combines recent ideas on set prediction (Locatello et al., 2020; Kosiorek et al., 2020). We concatenate the pixels of the input image with the camera position $\mathbf{x_0}$, and a positional encoding (Mildenhall et al., 2020) of the direction $\mathbf{d}$ of the corresponding rays. This is encoded into a (potentially smaller) feature map $\mathbf{y}$. We initialize the slots $z_i$ by sampling from a Gaussian distribution with learnable parameters. Following Locatello et al. (2020), we apply a few iterations of cross-attention between the slots and the elements of the feature map, interleaved with self-attention between the slots (similar to Kosiorek et al. (2020)). These two phases allow the slots to take responsibility for explaining parts of the input, and facilitate better coordination between the slots, respectively. The number of iterations is fixed, and all parameters are shared between iterations. The resulting slots form the latent representation of the scene. We note that it would be straightforward to use multiple views as input to the encoder, but we have not tried that in this work. For further details, we refer the reader to Appendix C.1.

**Decoder.** The decoder largely follows the MLP architecture of Mildenhall et al. (2020): We pass the Fourier-encoded inputs $\mathbf{x}$ through a series of fully-connected layers, eventually outputting the density $\sigma(\mathbf{x})$ and a hidden code $\mathbf{h}$. To condition the MLP on a latent code $z_i$, we use the code to shift and scale the activations at each hidden layer, a technique which resembles AIN of (Dumoulin et al., 2017; Brock et al., 2019). The color value $\mathbf{c}(\mathbf{x}, \mathbf{d})$ is predicted using two additional layers from $\mathbf{h}, \mathbf{d}$, and a pixel in the feature map $\mathbf{y}$ selected by projecting the query point $\mathbf{x}$ onto it. This conditioning follows (Yu et al., 2021a), except that we only use it to predict color, in order to not compromise the idea of the slots encoding the segmented geometry. We explored using $z_i$ to predict an explicit linear coordinate transform between the object and the world space as in (Niemeyer & Geiger, 2020; Elich et al., 2021), but have not found this to be beneficial for performance. We also note that we deliberately choose to decode each slot $z_i$ independently. This means we can treat objects as independent volumes, and in particular we can render one object at a time. Computing interactions between slots during decoding using e.g. attention (as proposed by Kosiorek et al. (2021)) would likely improve the model's ability to output complex visual features such as reflections, but would give up these benefits. Further details are described in Appendix C.2.

| Model | Sprites (bin) | Sprites | CLEVR |
|---|---|---|---|
| ObSuRF | $\mathbf{74.4 \pm 1.8}$ | $\mathbf{92.4 \pm 1.3}$ | $98.3 \pm 0.8$ |
| sel. runs* | $74.4 \pm 1.8$ | $93.1 \pm 0.3$ | $99.0 \pm 0.0$ |
| SlotAtt. | $69.4 \pm 0.9$ | $91.3 \pm 0.3$ | $\mathbf{98.8 \pm 0.3}$ |
| IODINE | $64.8 \pm 17.2$ | $76.7 \pm 5.6$ | $\mathbf{98.8 \pm 0.0}$ |
| MONet | - | $90.4 \pm 0.8$ | $96.2 \pm 0.6$ |
| R-NEM | $68.5 \pm 1.7$ | - | - |

Table 1: Average foreground ARI on 2D datasets (in %, mean $\pm$ standard deviation across 5 runs, the higher the better), compared with values reported in the literature (Locatello et al., 2020; Greff et al., 2020; Burgess et al., 2019; van Steenkiste et al., 2018). Best values are bold.

(*) For one run on Sprites and 2 runs on CLEVR, ObSuRF learns to segment background from foreground. While desirable, this slightly reduces the foreground ARI scores as extracting exact object outlines is more difficult. To quantify this, we also report results with these runs excluded.

**Training.** The model uses a minibatch of scenes at every training iteration. For every scene, we use one image (and its associated camera pose, but without depth) as input to the encoder. This yields the latent state $\{z_i\}_{i=1}^n$. We then average the loss on a random subset of rays sampled from the available RGB-D views of that scene. Finally, we average across the minibatch.

One advantage of 3D representations is that they allow us to explicitly express the prior knowledge that objects should not overlap. This is not possible in 2D, where one cannot distinguish between occluding and intersecting objects. We enforce this prior by adding the *overlap* loss $\mathcal{L}_O(\mathbf{r}) = \sum_i \sigma_i(\mathbf{r}(t)) - \max_i \sigma_i(\mathbf{r}(t))$, optimizing the overall loss $\mathcal{L} = -\log p(t, C) + k_O \mathcal{L}_O(\mathbf{r})$. Experimentally, we find that $\mathcal{L}_O$ can prevent the model from learning object geometry at all when present from the beginning. We therefore start with $k_O = 0$ and slowly increase its value in the initial phases of training. In turn, we find that we do not need the learning rate warm-up of Locatello et al. (2020). We describe all hyperparameters used in Appendix E.

## 4 EXPERIMENTAL EVALUATION

ObSuRF is designed for unsupervised volumetric segmentation of multi-object 3D scenes. As there are no published baselines for this setting, we start by evaluating it on 2D images instead of 3D scenes. This allows us to gauge how inductive biases present in a NeRF-based decoder affect unsupervised segmentation. We also check how a slotted representation affects reconstruction quality (compared to a monolithic baseline). We then move on to the 3D setting, where we showcase our model on our novel 3D segmentation benchmarks; we also compare the reconstruction quality to the closest available baseline, NeRF-VAE (Kosiorek et al., 2021).

**Metrics.** We evaluate segmentation quality in 2D image space by comparing produced segmentations to ground-truth using the *Adjusted Rand Index* (ARI, Rand (1971); Hubert & Arabie (1985)). The ARI measures clustering similarity and is normalized such that random segmentations result in a score of 0, while perfect segmentations in a score of 1. In line with Locatello et al. (2020), we not only evaluate the full ARI, but also the ARI computed on the foreground pixels (Fg-ARI; according to the ground-truth). Note that achieving high Fg-ARI scores is much easier when the model is not attempting to segment the background, i.e. , to also get a high ARI score: This is because ignoring the background allows the model to segment the objects using rough outlines instead of sharp masks. To measure the visual quality of our models' reconstructions, we report the mean squared error (MSE) between the rendered images and the corresponding test images. To test the quality of learned geometry in the 3D setting we also measure the MSE between the depths obtained via NeRF rendering and the ground-truth depths in the foreground (Fg-Depth-MSE). We exclude the background as a concession to the NeRF-VAE baseline, as estimating the distance of the background in the kind of data we use can be difficult from visual cues alone.

### 4.1 UNSUPERVISED 2D OBJECT SEGMENTATION

We evaluate our model on three unsupervised image segmentation benchmarks from the multi-object datasets repository (Kabra et al., 2019): CLEVR, Multi-dSprites, and binarized Multi-dSprites. We compare against four recent models as state-of-the-art baselines: SlotAttention (Locatello et al., 2020), IODINE (Greff et al., 2020), MONet (Burgess et al., 2019), and R-NEM (van Steenkiste

et al., 2018). We match the experimental protocol established by Locatello et al. (2020): On dSprites, we train on the first 60k samples, on CLEVR, we select the first 70k scenes for training, and filter out all scenes with more than 6 objects. Following prior work, we test on the first 320 scenes of each validation set (Locatello et al., 2020; Greff et al., 2020) and process CLEVR images by center-cropping and resizing to $64 \times 64$.

To adapt our encoder to the 2D case, we apply the positional encoding introduced by Locatello et al. (2020), instead of providing camera position and ray directions. To reconstruct 2D images, we query the decoder on a fixed grid of 2D points, and again drop the conditioning on viewing direction. During training, we add a small amount of Gaussian noise to this grid to avoid overfitting to its exact position. Similar to previous methods (Locatello et al., 2020; Greff et al., 2020), we combine the 2D images which the decoder produces for each slot by interpreting its density outputs $\sigma_i$ as the weights of a mixture, i.e. , we use the formula $\hat{C}(\mathbf{r}) = \sum_i c_i(\mathbf{r})\sigma_i/\sigma$. As we show in Appendix B.3, this is equivalent to shooting a ray through a superposition of volumes with constant densities $\sigma_i$ and infinite depth. We train the model by optimizing the color reconstruction loss.

In Table 1, we compare the foreground ARI scores achieved by our model with those from the literature. Visualizations are given in Appendix A.1. We find that our model performs significantly better on the Sprites data, especially the challenging binary variant, indicating that our decoder architecture imparts useful spatial priors. On CLEVR, our model largely matches the already nearly perfect results obtained by prior models. We note that like (Locatello et al., 2020), our model occasionally learns to segment the background in a separate slot, which is generally desirable but slightly reduces the Fg-ARI score as discussed above. Additionally, as we report in A.2, we have found ObSuRF to produce better reconstructions than an ablation with a monolithic latent code instead of a slot based representation, confirming that the latter is also helpful for prediction quality.

Training ObSuRF on these 2D datasets takes about 34 hours on a single V100 GPU. This is roughly 4 times faster than SlotAttention (133 hours on one V100 (Locatello et al., 2020)) and almost 40 times more efficient than IODINE (One week on 8 V100s (Greff et al., 2020)). We attribute the efficiency gains compared to SlotAttention to our decoder architecture, which processes each pixel individually via a small MLP instead of using convolutional layers.

## 4.2 UNSUPERVISED 3D OBJECT SEGMENTATION

To test ObSuRF's capability of learning volumetric object segmentations in 3D, we assemble two novel benchmarks. First, we adapt the CLEVR dataset (Johnson et al., 2017) from (Kabra et al., 2019) by rendering each scene from three viewpoints, and also collecting depth values. The initial viewpoint is the same as in the original dataset, the others are obtained by rotating the camera by $120°/240°$ around the $z$-axis. As in the 2D case, we restrict ourselves to scenes with at most 6 objects in them. Second, we construct MultiShapeNet, a much more challenging dataset which is structurally similar to CLEVR. However, instead of simple geometric shapes, each scene is populated by 2-4 objects from the ShapeNetV2 (Chang et al., 2015) 3D model dataset. The resulting MultiShapeNet dataset contains $11\,733$ unique shapes. We describe both datasets in detail in Appendix D. In contrast to the 2D case, we utilize the full $320 \times 240$ resolution for both datasets. Due to our decoder architecture, we do not need to explicitly generate full size images during training, which makes moving to higher resolution more feasible.

When testing our model, we provide a single RGB view of a scene from the validation set to the encoder. We can then render the scene from arbitrary viewpoints by estimating $\hat{C}(\mathbf{r})$ (Eq. (6)) via hierarchical sampling as in (Mildenhall et al., 2020). We set the maximum length $t_{\text{far}}$ of each ray to 40 or the point at which it intersects $z = -0.1$, whichever is smaller. This way, we avoid querying the model underneath the ground plane, where it has not been trained and where we cannot expect sensible output. We use a rendering obtained from the viewpoint of the input to compute the reconstruction MSE. In order to obtain depthmaps and compute the depth-MSE, we use the available samples to estimate the expected value of the depth distribution $\mathbb{E}_{p(t)}[t] = \int_0^{t_{\text{far}}} t\,p(t)dt$. We draw segmentation masks by computing $\arg\max p(i)$. Finally, we also render individual slots by applying the same techniques to the individual NeRFs, following Eqs. (2) and (3). In order to highlight which space they do and do not occupy, we use the probability $p(t \leq t_{\text{far}})$ as an alpha channel, and draw the slots in front of a checkerboard background.

| Model | CLEVR-3D | | | | MultiShapeNet | | | |
|---|---|---|---|---|---|---|---|---|
| | C-MSE | D-MSE | Fg-ARI | ARI | C-MSE | D-MSE | Fg-ARI | ARI |
| ObSuRF | **0.31** | **0.07** | 96.9 | 92.0 | **0.60** | **1.28** | 94.7 | 61.1 |
| without $\mathcal{L}_O$ | 0.60 | 0.25 | 94.0 | 1.4 | 0.82 | 3.32 | **94.9** | 16.9 |
| w/o px. cond. | 0.78 | 0.10 | 95.7 | **94.6** | 1.81 | 3.44 | 81.4 | **64.1** |
| w/o px. cond., $\mathcal{L}_O$ | 0.80 | 0.12 | 85.5 | 4.83 | 1.78 | 3.40 | 94.4 | 16.5 |
| w/ NeRF loss | 0.69 | 0.25 | 97.8 | 18.0 | 2.08 | 4.00 | 58.75 | 3.1 |
| w/ D-NeRF loss | 1.11 | 0.15 | **97.9** | 15.7 | 4.7 | 8.95 | 12.8 | 0.2 |
| NeRF-VAE | 4.7 | 1.03 | - | - | 5.5 | 110.3 | - | - |

Table 2: Quantitative results on the 3D datasets. C-MSE scores are colour MSE $\times 10^3$; D-MSE are depth MSE values computed on the foreground only. ARI scores are in percent, best values in bold. $\mathcal{L}_O$ is the overlap loss.

We report results for the full ObSuRF model, which uses both the overlap loss $\mathcal{L}_O$ and pixel conditioning, and a number of ablations. These do not use the overlap loss (w/o $\mathcal{L}_O$), or pixel conditioning (w/o pixel cond.). Additionally, we report results for ObSuRF trained with vanilla NeRF rendering (w/ NeRF loss) and with NeRF rendering plus depth loss of (Deng et al., 2021) (w/ D-NeRF loss). Details on these ablations are given in Appendix C.5. The slot sizes are 128 for CLEVR-3D and 256 for MultiShapeNet. As a baseline, we compare to NeRF-VAE which, similarly to our model, is an autoencoder where the NeRF decoder is conditioned on a latent extracted from a single input image. However, it is trained without depth information, and does not segment the scene. Instead, it maximizes the evidence lower-bound (ELBO), comprised of a reconstruction term and a regularizing KL-divergence term. It also uses much larger latent representations of size $8 \times 8 \times 128$, and an attention-based decoder. While the differences in architecture make training times impossible to directly compare, we estimate that due to our RGB-D based loss, ObSuRF required $24\times$ fewer evaluations of its scene function than NeRF-VAE over the course of training.

Figure 1 and Table 2 contain qualitative and quantitative results, respectively. We find that ObSuRF learns to segment the scenes into objects, and to accurately reconstruct their position, dimensions, and color. On the MultiShapeNet dataset, the large variety among objects causes the model to sometimes output vague volumes instead of sharp geometry, whereas on CLEVR-3D, it stays close to the true scene. We observe that similarly to the 2D models, our model does not learn to segment the background into its own slot without $\mathcal{L}_O$. With this additional loss term however, it learns to also segment the background, leading to much higher ARI scores. As expected, pixel conditioning increases the accuracy of predicted colors and geometry. On CLEVR3D, both versions of NeRF training yields similar results to our loss, whereas on MultiShapeNet, they fail to produce accurate segmentations or geometry, with depth supervision Deng et al. (2021) actually performing worse. In this more challenging setting, we also find that our training method converges much faster, as shown by the learning curves in Appendix A.3. Compared to NeRF-VAE, we find that our model achieves much lower reconstruction errors on both datasets.

## 5 CONCLUSION

We present ObSuRF, a model that segments 3D scenes into objects represented as NeRFs. We have shown that it learns to infer object positions, dimensions, and appearance on two challenging and novel 3D modelling benchmarks. Importantly, it does so using purely observational data, without requiring supervision on the identity, geometry or position of individual objects.

Since ObSuRF learns to infer compact object representations that *have to* be well informed of their geometry (due to the model structure), using these representations may be beneficial for downstream tasks including robot grasping, dynamics modelling, or visual question answering. A key step in this direction is to improve inference capabilities and robustness to noisy depth supervision, which will enable working with real-world data.

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

| Model | Sprites (bin) | Sprites | CLEVR |
|-------|---------------|---------|-------|
| ObSuRF | $1.34 \pm 0.15$ | $0.64 \pm 0.11$ | $0.41 \pm 0.06$ |
| NeRF-AE | 4.82 | 2.60 | 1.75 |

Table 3: Testset reconstruction MSEs $\times 10^3$ of our model compared to the NeRF-AE ablation, which uses only a single latent vector.

Zachary Teed and Jia Deng. Deepv2d: Video to depth with differentiable structure from motion. In *ICLR*, 2020. 2

Alex Trevithick and Bo Yang. Grf: Learning a general radiance field for 3d scene representation and rendering. In *arXiv:2010.04595*, 2020. 3

Sjoerd van Steenkiste, Michael Chang, Klaus Greff, and Jürgen Schmidhuber. Relational neural expectation maximization: Unsupervised discovery of objects and their interactions. In *ICLR*, 2018. 3, 7

Nanyang Wang, Yinda Zhang, Zhuwen Li, Yanwei Fu, Wei Liu, and Yu-Gang Jiang. Pixel2mesh: Generating 3d mesh models from single rgb images. In *ECCV*, 2018. 1

Marissa A. Weis, Kashyap Chitta, Yash Sharma, Wieland Brendel, Matthias Bethge, Andreas Geiger, and Alexander S. Ecker. Unmasking the inductive biases of unsupervised object representations for video sequences. In *arXiv:2006.07034*, 2020. 1, 3

Alex Yu, Vickie Ye, Matthew Tancik, and Angjoo Kanazawa. pixelNeRF: Neural radiance fields from one or few images. In *CVPR*, 2021a. 3, 6, 17

Hong-Xing Yu, Leonidas J. Guibas, and Jiajun Wu. Unsupervised discovery of object radiance fields. In *arXiv:2107.07905*, 2021b. 3

Yan Zhang, Jonathon Hare, and Adam Prügel-Bennett. Fspool: Learning set representations with featurewise sort pooling. In *ICLR*, 2020. 3

## A ADDITIONAL RESULTS

### A.1 2D VISUALIZATIONS

In Fig. 3, we provide visualizations of ObSuRF's 2D results.

### A.2 2D ABLATION

To quantify the impact of learning representations which are segmented into slots, we construct a baseline which uses only a single slot, called NeRF-AE. To keep the total computational capacity comparable to our model, we set the size of this single latent vector to be equal to the total size of the slots ObSuRF is using. We also double the size of the hidden layers in encoder and decoder. Still, as we show in Table 3, our model consistently achieves much lower reconstruction errors. This confirms previous results indicating that representing multi-object scenes using a set of permutation equivariant elements is not just helpful for segmentation, but also for prediction quality (Kosiorek et al., 2021; Locatello et al., 2020).

### A.3 RUNTIME COMPARISON

We show learning curves of ObSuRF (without $\mathcal{L}_O$) and the NeRF based ablations in Figure 4. While the models behave comparably on CLEVR3D, ObSuRF converges much faster and to a much better segmentation result on Shapenet. Training time on the $x$-axis is measured in scene function evaluations, as these make up at least 99% of the computational cost of training each model.

## B PROOFS AND DERIVATIONS

We start by providing detailed derivations for the results referenced in the main text.

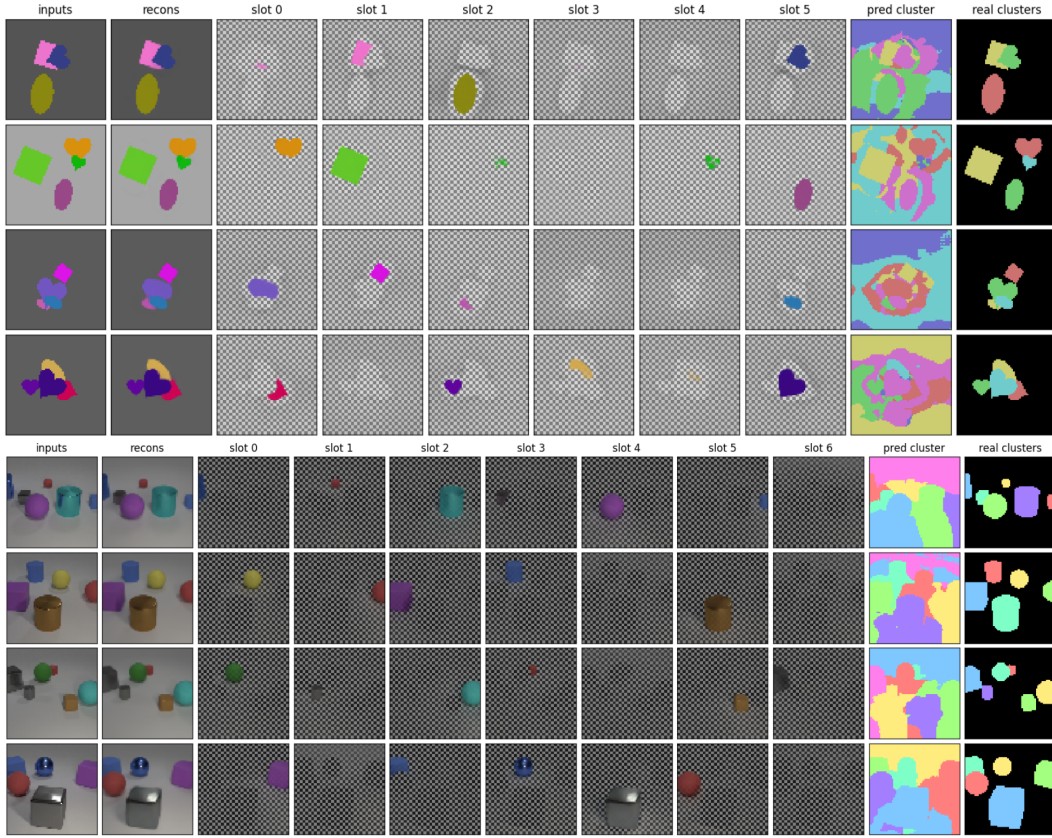

Figure 3: Visualisations of ObSuRF's output given samples from the validation set of Sprites (top) and CLEVR2D (bottom).

## B.1 DERIVATION OF THE NERF DEPTH DISTRIBUTION

In Section 3.1, we noted that the distribution $p(t)$ shown in Eq. (2) arises both as the distribution of points along $\mathbf{r}$ who contribute to the color observed at $\mathbf{x_0} = \mathbf{r}(0)$, and as the distribution of points illuminated by a light source located at $\mathbf{x_0}$. Here, we start by showing the latter.

### B.1.1 DISTRIBUTION OF LIGHT ORIGINATING AT $\mathbf{x_0}$

Following the low albedo approximation due to Blinn (1982), we ignore indirect lighting. In order for light originating at $\mathbf{x_0}$ to illuminate $\mathbf{r}(t)$ directly, it must travel along the ray without any scattering events. The rate of scattering events is given by the inhomogenous spatial Poisson process with finite, non-negative density $\sigma(\mathbf{x})$ (Møller & Waagepetersen, 2003). We are therefore looking for the distribution over the position of the first event encountered while traveling along $\mathbf{r}$, also called the *arrival* distribution of the process (Møller & Waagepetersen, 2003). In a Poisson process, the probability of encountering no events along a given trajectory $\mathbf{r}(t)$ with $t \geq 0$ (transmittance) is

$$T(t) = \exp\left(\int_0^t -\sigma(\mathbf{r}(t'))dt'\right). \tag{7}$$

The probability that light scatters before moving past some point $t_0$ is therefore $p(t \leq t_0) = 1 - T(t_0)$. Let us now assume that $\lim_{t_0 \to \infty} T(t_0) = 0$, i.e., each ray of light encounters a scattering event eventually. This is for instance guaranteed if the scene has solid background. In that case, $1 - T(t)$ is the cumulative distribution function for $p(t)$, since it is non-decreasing, continuous, and

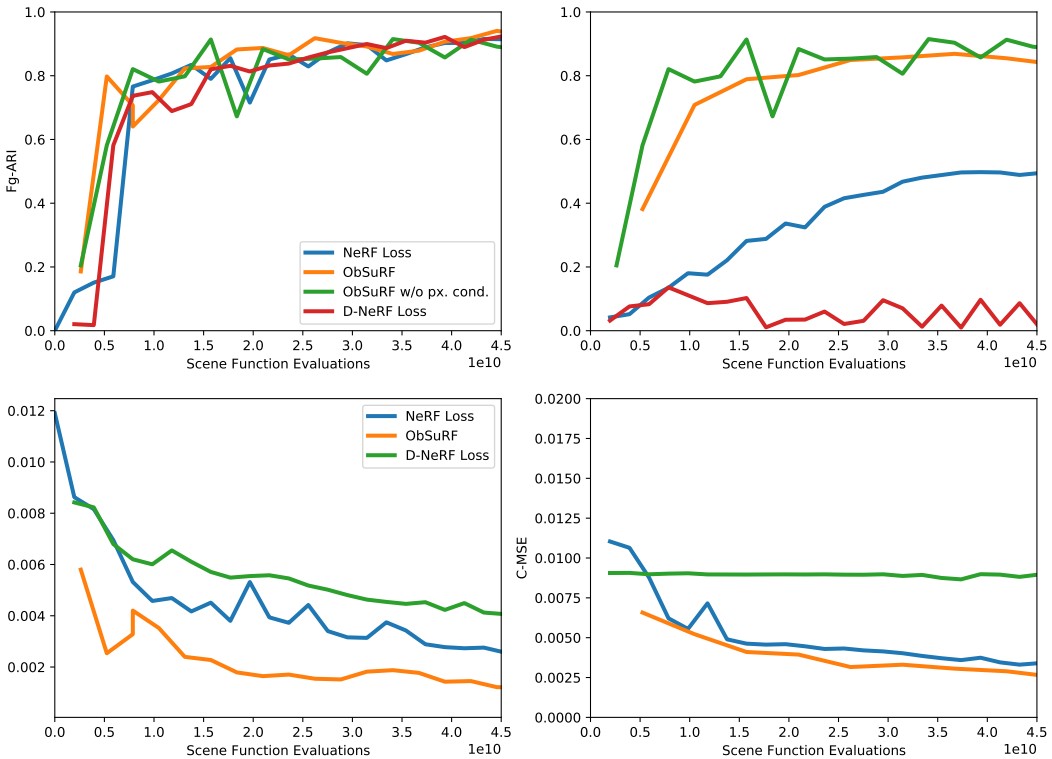

Figure 4: Learning curves of ObSuRF and its ablations measuring Fg-ARI (top) and C-MSE (bottom) when training on CLEVR3D (left) and Multishapenet (right). For better comparability with NeRF-based training, all ObSuRF models are trained without the overlap loss $\mathcal{L}_O$.

fulfills $T(0) = 0, T(\infty) = 1$. We can therefore recover the density function $\mathrm{p}(t)$ by differentiating:

$$\mathrm{p}(t) = \frac{d}{dt}\left(1 - T(t)\right) \tag{8}$$

$$= -T(t)\frac{d}{dt}\int_0^t -\sigma(\mathbf{r}(t'))dt' \tag{9}$$

$$= \sigma(\mathbf{r}(t))T(t). \tag{10}$$

### B.1.2 DISTRIBUTION OF COLORS OBSERVED AT $\mathbf{x_0}$

As discussed in Section 3.1, NeRF does not model light sources explicitly, but assumes that lighting conditions are expressed by each point's RGB color value. The observed color $\hat{C}(\mathbf{r})$ at point $\mathbf{x_0}$ along ray $\mathbf{r}$ is then a blend of the color values along $\mathbf{r}(t)$ (Blinn, 1982; Kajiya & von Herzen, 1984). The intensity with which we can observe the color emitted from point $\mathbf{r}(t)$ in direction $-\mathbf{d}$ of the camera is proportional to the density $\sigma(\mathbf{r}(t))$. Let $k$ denote the constant of proportionality. Since $T(t)$ is the probability that light reaches $\mathbf{x_0}$ from $\mathbf{r}(t)$, the we can observe the color $\mathbf{c}(\mathbf{r}(t), \mathbf{d})$ at $\mathbf{x_0}$ with intensity $k\sigma(\mathbf{r}(t))T(t)$.

In order to obtain the distribution $\mathrm{p}(t)$ over the points from which the colors we observe at $\mathbf{x_0}$ originate, we must normalize this term by dividing by the *total* intensity along ray $\mathbf{r}$: $\mathrm{p}(t) =$

$$\frac{k\sigma(\mathbf{r}(t))T(t)}{\int_0^\infty k\sigma(\mathbf{r}(t'))T(t')dt'} = \frac{\sigma(\mathbf{r}(t))T(t)}{\int_0^\infty \sigma(\mathbf{r}(t'))T(t')dt'}. \tag{11}$$

As we have shown in Eq. (10), the antiderivative of $\sigma(\mathbf{r}(t))T(t)$ is $-T(t)$. The value of the improper integral is therefore

$$\lim_{t\to\infty}\int_0^t \sigma(\mathbf{r}(t'))T(t')dt' = \lim_{t\to\infty} T(0) - T(t) = 1. \tag{12}$$

Consequently, $p(t) = \sigma(\mathbf{r}(t))T(t)$. This procedure may be viewed as a continuous version of alpha compositing (Porter & Duff, 1984).

## B.2 ESTIMATING $\mathcal{L}_{\text{NeRF}}$ IS BIASED

As a counterexample, consider the following situation in which a ray $\mathbf{r}$ passes through a thin but dense white volume, before hitting black background. For simplicity, we use grayscale color values.

$$\sigma(\mathbf{r}(t)) = \begin{cases} 100 & \text{if } 50 \leq t \leq 51 \\ 10 & \text{if } t > 80 \\ 0 & \text{otherwise,} \end{cases} \tag{13}$$

$$c(\mathbf{r}(t), \mathbf{d}) = \begin{cases} 1 & \text{if } 50 \leq t \leq 51 \\ 0 & \text{otherwise.} \end{cases} \tag{14}$$

We then have $T(50) = 1$, but $T(51) = T(80) = \exp(-100)$. Since $T(\cdot)$ is the CDF of $t$ (Eq. (10)), this means almost all of the incoming light is coming from the thin volume at $t \in [50, 51]$:

$$p(50 \leq t \leq 51) = 1 - \exp(-100) \tag{15}$$
$$p(t > 80) = \exp(-100). \tag{16}$$

Using Eq. (3), we find that the color value is almost exactly one:

$$\hat{C}(\mathbf{r}) = 1 - \exp(-100) \approx 1. \tag{17}$$

Let us now consider the result of estimating $\hat{C}(\mathbf{r})$ using $k = 50$ stratified samples $t_1, \ldots, t_k$, with $t_{\text{far}} = 100$. In this case, one sample $t_i$ is drawn uniformly from each of $k$ evenly sized bins:

$$t_i \sim \text{Uniform}\left(\frac{i-1}{k}t_{\text{far}}, \frac{i}{k}t_{\text{far}}\right). \tag{18}$$

Only one of the bins, the one for $t_{26} \sim \text{Uniform}(50, 52)$, covers the volume at $[50, 51]$. However, since this volume only makes up half the range of this bin, we have $p(50 \leq t_{26} \leq 51) = 1/2$. Therefore, 50% of the time, sampling in this way will result in samples which miss the volume at $[50, 51]$ entirely. In those cases, we have $c(\mathbf{r}(t_i), \mathbf{d}) = 0$ for all $i$. As a result, the estimated color value is $\hat{C}^{|t_i}(\mathbf{r}) = 0$. Even if the estimated color is 1 in all other cases, this sampling scheme will be a biased estimator for the true color values:

$$\mathbb{E}_{t_i}\left[\hat{C}^{|t_i}(\mathbf{r})\right] \leq \frac{1}{2} < \hat{C}(\mathbf{r}). \tag{19}$$

Collecting a second round of samples based on the estimated densities as proposed by Mildenhall et al. (2020) does not change the result. In the 50% of cases where the thin volume was missed entirely during by the first set of samples, we have $\sigma(\mathbf{r}(t_i)) = 0$ for all $t_i \leq 80$. As a result, the piecewise-constant PDF used for sampling the second round of samples will equal 0 for $50 \leq t \leq 51$, and none of the newly collected samples will cover the relevant range. Therefore, Eq. (19) also holds when the samples $t_i$ were collected using hierarchical sampling.

We note that this effect may be somewhat mitigated by the fact that NeRF models typically use different scene functions for coarse and fine grained sampling (Mildenhall et al., 2020; Kosiorek et al., 2021). The coarse scene function may then learn to model wider volumes which are easier to discover while sampling, instead of the exact scene geometry. However, it seems clear that this will not allow decreasing the number of samples arbitrarily, e.g. to 2 as we did in this paper.

## B.3 SUPERPOSITION OF CONSTANT DENSITIES YIELDS A MIXTURE MODEL

Here, we illustrate how a mixture model, like the one we use in Section 4.1, naturally arises when a superposition of volumes with constant densities $\sigma_i$ and colors $c_i$ is rendered. Following Eq. (5) and writing $\sigma = \sum_i \sigma_i$, the transmittance along a ray $\mathbf{r}$ is

$$T(t) = \exp\left(-\int_0^t \sigma dt'\right) = \exp(-\sigma t). \tag{20}$$

As in Section 3.4, we can then write

$$p(t, i) = \sigma_i(\mathbf{r}(t))T(t) = \sigma_i \exp(-\sigma t). \qquad (21)$$

Integrating over $t$ to obtain the distribution over the volumes $i$ yields

$$p(i) = \int_0^\infty p(t, i)dt = \left[ \frac{\sigma_i}{-\sigma} \exp(-\sigma t) \right]_0^\infty = \frac{\sigma_i}{\sigma}. \qquad (22)$$

Using $c_i(\mathbf{x}) = c_i$, and following Eq. (6), we find that the observed color will then be

$$\mathbb{E}_{i \sim p(\cdot)}[c_i] = \sum_{i=1}^n c_i \frac{\sigma_i}{\sigma}, \qquad (23)$$

which is the desired mixture model.

## C ARCHITECTURES

We now present the architectures used in our experiments. When we refer to some of the dimensions in terms of a variable, we specify the concrete values used in Appendix E. ObSuRF uses exclusively ReLU activations.

### C.1 ENCODER ARCHITECTURE

Due to the differences in resolution and complexity, we used different feature extractor architectures for the 2D and 3D experiments. The slot attention module was identical in both cases.

**Feature Extractor for 2D Datasets.** Following Locatello et al. (2020), we use a convolutional network with 4 layers to encode the input image for the 2D datasets. Each layer has a kernel size of $5 \times 5$, and padding of 2, keeping the spatial dimensions of the feature maps constant at $64 \times 64$. Each layer outputs a feature map with $d_h$ channels. After the last layer, we apply the spatial encoding introduced by Locatello et al. (2020). Again following Locatello et al. (2020), we process each pixel in the resulting feature map individually using a layer normalization step, followed by two fully connected (FC) layers with output sizes $[d_h, d_z]$.

**Feature Extractor for 3D Datasets.** For the 3D experiments, we adapt the ResNet-18 architecture to achieve a higher model capacity. We start by appending the camera position and a positional encoding (Mildenhall et al., 2020) of the viewing direction to each pixel of the input image, and increase the input size of the ResNet accordingly. We remove all downsampling steps from the ResNet except for the first and the third, and keep the number of output channels of each layer constant at $d_h$. Given an input image of size $240 \times 320$, the extracted feature map therefore has spatial dimensions $60 \times 80$, and $d_h$ channels. We apply the same pixelwise steps described for the 2D case (layer normalization followed by two FC layers) to obtain the final feature map with $d_z$ channels.

**Slot Attention.** We apply slot attention as described by Locatello et al. (2020) with $n$ slots for $m$ iterations, except for the following change. At each iteration, after the GRU layer, we insert a multi-head self attention step with 4 heads to facilitate better coordination between the slots. Specifically, we update the current value of the slots via

$$\mathbf{z} := \mathbf{z} + \text{MultiHead}(\mathbf{z}, \mathbf{z}, \mathbf{z}). \qquad (24)$$

After $m$ iterations, the values of the slots form our latent representation $z_1, \ldots z_n$. We apply the same number of iterations $m$ during both training and testing.

### C.2 DECODER ARCHITECTURE

Here, we describe how density and colors are predicted from the spatial coordinates $\mathbf{x}$ and a conditioning vector $z$. If pixel conditioning is not used, then $z$ is simply equal to one of the latent codes $z_i$. Otherwise, we concatenate $z_i$ with the feature map pixel $d_z^{r,c}$, where $r, c$ are the image space coordinates of the query point $\mathbf{x}$ projected onto the feature map $d_z$, following Yu et al. (2021a).

To then predict color and density, we first encode the $\mathbf{x}$ via positional encoding (Mildenhall et al., 2020), using $n_f$ frequencies, with the lowest one equal to $2^{k_f}\pi$. We process the encoded positions

| Parameter | Description | 2D Data | CLEVR-3D | MultiShapeNet |
|---|---|---|---|---|
| $d_z$ | Dimensionality of slots | 64 | 128 | 256 |
| $d_h$ | Dimensionality of hidden layers | 64 | 128 | 256 |
| $m$ | Number of Slot Attention iterations | 3 | 5 | 5 |
| $n_f$ | Number of frequencies for pos. enc. | 8 | 16 | 16 |
| $k_f$ | Exponent for lowest frequency | 0 | -5 | -5 |
| $\sigma_{\max}$ | Maximum density | - | 10 | 10 |
| $\sigma_c$ | Standard deviation of color dist. | 1 | 0.2 | 0.2 |
| $\delta$ | Noise added to depths $t$ | - | 0.07 | 0.07 |
| $\hat{k}_O$ | Maximum overlap loss coefficient | - | 0.05 | 0.03 |
| | Start of $\mathcal{L}_O$ ramp up | - | 20000 | 20000 |
| | End of $\mathcal{L}_O$ ramp up | - | 40000 | 40000 |
| | Batch Size | 128 | 64 | 64 |
| | Rays per instance | - | 4096 | 2048 |

Table 4: Hyperparameters used for the experiments with ObSuRF.

using a 5 layer MLP with hidden size $d_h$. After each hidden layer, we scale and shift the current hidden vector $h$ elementwise based on $z$. Specifically, at each layer, we use a learned linear map to predict scale and shift parameters $\alpha, \beta$ (each of size $d_h$) from $z$, and update $\mathbf{h}$ via $\mathbf{h} := (\mathbf{h} + \beta) \cdot \alpha$.

In the 2D case, we output 4 values at the last layer, namely the log density $\log \sigma$ and the RGB-coded colors $\mathbf{c}$. In the 3D case, we output one value for the density, and $d_h$ values to condition the colors. We append the view direction $\mathbf{d}$ to the latter, and predict the actual color output using two additional FC layers of size $[d_h, 3]$, followed by a sigmoid activation. In the 3D case, we find that it is necessary to set a maximum density value, as the model can otherwise arbitrarily inflate the depth likelihood $p(t)$ by increasing the density in regions which are known to always be opaque, such as the ground plane. We do this by applying a sigmoid activation to the decoder's density output, and multiplying the result by $\sigma_{\max}$, obtaining values in the range $[0, \sigma_{\max}]$.

### C.3 NeRF-AE Baseline Details

For the NeRF-AE baseline (Table 3), we adapt our encoder architecture to output a single vector instead of a feature map. To this end, we first append a positional encoding (Mildenhall et al., 2020) of each pixel's coordinates to the input image. This replaces the spatial encoding described above, since the method described by Locatello et al. (2020) requires a full-sized feature map to be applicable. We double the number of channels to $2d_h$, but set the convolutional layers to use stride 2, obtaining an output with spatial dimensions $4 \times 4$. As above, each of these vectors is processed via layer normalization and two fully connected layers, this time with output sizes $[2d_h, 2d_h]$. We flatten the result to obtain a vector of size $32 \cdot d_h$, and obtain the final vector using an MLP of sizes $[4d'_z, 2d'_z, d'_z]$. This replaces the slot attention component described below. To keep the total dimensionality of the latent space identical, we set $d'_z = nd_z$.

### C.4 NeRF-VAE Baseline Details

We compare ObSuRF to NeRF-VAE on the quality of image reconstruction and depth estimation. We use NeRF-VAE with the attentive scene function and without iterative inference as reported by Kosiorek et al. (2021), with the majority of parameters the same. The only differences stem from different image resolution used in the current work ($240 \times 320$ as opposed to $64 \times 64$ in (Kosiorek et al., 2021)). To accommodate bigger images, we use larger encoder with more downsampling steps. Specifically, we use [(3, 2), (3, 2), (3, 2), (3, 2) (4, 1)] block groups as opposed to [(3, 2), 3, 2), (4, 1)], which results in an additional $4\times$ downsampling and 6 additional residual blocks. We used the same parameters to train the NeRF-VAE on both datasets: Models were trained for $2.5 \times 10^5$ iterations with batch size 192, learning rate of $5 \times 10^{-4}$ and Adam optimizer. The training objective is to maximize the ELBO, which involves reconstructing an image with volumetric rendering. We sample 32 points from the coarse scene net, and we reuse those points plus sample additional 64 points for evaluating the fine scene net. Kosiorek et al. (2021) were able to set the integration cutoff

$t_{\text{far}} = 15$, but for the data generated for this paper, this value was too small and we used $t_{\text{far}} = 40$. This effectively reduces the number of points sampled per unit interval, which might explain higher reconstruction and depth-estimation errors.

### C.5 NeRF Training Ablation Details

The ablations involving training via NeRF-style ray marching largely use the same architecture as ObSuRF, except for the following changes:

- No pixel conditioning is used, as we have found that this causes the model to fail to segment, likely because it allows the model to achieve good loss scores without relying on the slots.

- No type of overlap loss is used.

- The maximum density is not limited to a maximum value, instead it is predicted using a simple ReLU activation.

- Following Mildenhall et al. (2020), we learn two instances of the decoder for the coarse and fine rendering passes, respectively.

- For training on CLEVR3D, we collect 48 samples each for the coarse and fine rendering pass. For shapenet, we use 64 samples each. To account for the increased memory cost, we reduce the batch size to 16, and the number of rays per instance to 256 and 192, respectively.

- For depth supervision, an MSE loss was placed on the estimated depths. Following Deng et al. (2021), this loss was weighted with the factor $\lambda = 0.1$ for CLEVR3D. For MultiShapeNet, we tried $0.1$ and $0.04$, but found that neither causes the model to properly segment the scene. The results above were obtained using $\lambda = 0.04$.

## D  Datasets

Here, we describe both of the 3D datasets used.

### D.1  CLEVR-3D

To allow for maximum interoperability with the 2D version of CLEVR, we use the scene metadata for the CLEVR dataset provided by Kabra et al. (2019). While this data does not specify the original (randomized) camera positions, we were able to recover them based on the provided object coordinates in camera space via numerical root finding. We were therefore able to exactly reconstruct the scenes in Blender, except for the light positions, which were rerandomized. To ensure a coherent background in all directions, we added a copy of the backdrop used by CLEVR, facing in the opposite direction. We rendered RGB-D views of the scene from the original camera positions and two additional positions obtained by rotating the camera by $120°/240°$ around the z axis. We test on the first 500 scenes of the validation set, using RGB images from the original view points as input.

### D.2  MultiShapeNet

For the MultiShapeNet dataset, we start with the same camera, lighting, and background setup which was used for CLEVR-3D. For each scene, we first choose the number of objects uniformly between 2 and 4. We then insert objects one by one. For each one, we first uniformly select one of the *chair*, *table*, or *cabinet* categories. We then uniformly choose a shape out of the training set of shapes provided for that category in the ShapeNetV2 dataset (Chang et al., 2015), leaving the possibility for future evaluations with unseen shapes. We insert the object at a random $x, y$ position in $[-2.9, 2.9]^2$, scale its size by a factor of 2.9, and choose its $z$ position such that the bottom of the object is aligned with $z = 0$. To prevent intersecting objects and reduce the probability of major occlusions, we compute the radius $r$ of the newly inserted object's bounding circle in the $xy$-plane. If a previously inserted object has radius $r'$, and their $xy$-distance $d < 1.1(r + r')$, we consider the objects to be too close, remove the last inserted object, and sample a new one. If the desired number of objects has not been reached after 20 iterations, we remove all objects and start over.

# E    MODEL PARAMETERS

We report the hyperparameters used for ObSuRF in Table 4. In addition to the architectural parameters introduced above, we note the importance of the standard deviation of the color distribution $\sigma_C$ for tuning the relative importance of color compared to depth. We also report how we ramp up the overlap loss $\mathcal{L}_O$: From the beginning of training to the iteration at which we start the ramp up, we set $k_O = 0$. During the ramp up period, we linearly increase $k_O$ until it reaches the maximum value $\hat{k}_O$. We train using the Adam optimizer with default parameters, and an initial learning rate of $4e - 4$. We reduce the learning rate by a factor of $0.5$ every 100k iterations. Finally, we note that for the 3D models, we used gradient norm clipping during training, i.e. , at each iteration, we determine the L2 norm of the gradients of our model as if they would form a single vector. If this norm exceeds 1, we divide all gradients by that norm. When training on MultiShapeNet, we find that very rare, extremely strong gradients can derail training. After the first 5000 iterations, we therefore skip all training steps with norm greater than 200 entirely.

