# OpenReview forum: "Decomposing 3D Scenes into Objects via Unsupervised Volume Segmentation"
_ICLR.cc/2022/Conference — ICLR 2022 Submitted_

### Official Review · Reviewer_VrrK · 2021-11-02

**Correctness:** 3
**Technical Novelty And Significance:** 2
**Empirical Novelty And Significance:** 2
**Recommendation:** 5
**Confidence:** 3

**Main Review:**

Strength:

- Decomposing a scene into objects in 3D is a challenging problem and solving this could potentially be useful for the area of explainable AI.

- The probabilistic interpretation of ray marching in Section 3.2 is interesting and somewhat novel in the context of NeRF, but this piece of knowledge is well understood in the physically based rendering community.

Weakness:
- The problem authors seek to address is not defined clearly. The title and the
main concept of the paper are inconsistent. According to the title, a new 3D scene segmentation method is proposed. However, the input of the pipeline is 2D images which is prone to be 2D instance segmentation.
- Lack of novelty.
Using depth to facilitate training is proposed[1].
[1]Depth-supervised NeRF: Fewer Views and Faster Training for Free
Composing NeRFs has been explored before, as acknowledged by the authors in the citations. This paper arrives at similar formulations but from the probabilistic point of view.

- The presentation sometimes gets complicated with too long background information (Section 3.1, 3.2, 3.3) that causes some confusions.

**Summary Of The Paper:**

The paper presents an array of techniques for solving the problem of decomposing a single image of a scene into a set of NeRFs, each of which represents an object in the scene. The decomposition requires accurate estimation of scene depth and the segmentation of the scene into objects. To achieve this, the authors propose an enhanced version of NeRF that operates on RGB-D images where its rendering only requires two evaluations per each camera ray. The authors show that the proposed method works effectively on CLEVR-3D and a custom built dataset named MultiShapeNet.


**Summary Of The Review:**

Summary
The authors use an encoder network to infer a set of slots for each object contained in the input image and then these features are used in Nerf to generate novel views. However, it lacks the motivation of leveraging nerf lack. The authors must make it clear. In addition, it lacks experiments to verify the proposed methods. Evaluation of advanced 3D object dataset such as Google Scanned Objects is necessary. Overall, the paper should be rejected.

Questions to authors

1) The paper seems to include some contributions that do not directly solve the problem. For example, can the decomposition work with traditional NeRF and ray marching instead of the proposed ray evaluation?

2) Section 3.3 mentions that the ray marching in NeRF is biased, which is true and a well known phenomenon in volume rendering.
Then the authors claimed that the proposed ray evaluation is unbiased. Could you provide a formal proof?

3) The comparison does not include results that evaluates segmentation and depth separately using state-of-the-art methods for each aspect. How does this method fare compared to those SOTAs?

4) Ablation study is lacking. Overall I can see that the proposed method outperforms a few baselines but there is not much analysis into the proposed method. Statistics about running time should be shown too as the proposed ray evaluation could be faster than traditional ray marching in NeRF.
For contribution in Section 3.3., output quality compared to NeRF should also be shown.

5) The experiments are done with synthetic datasets only. Since the proposed method is based on NeRF, how well does it work with real photographs?

Detailed comments

- Abstract: the motivation of segmentation is not clear. Infering a set of NeRFs, one for each object, from a single input image is a good approach, but the connection to segmentation of the scene is left unexplained.

- The derivations are sometimes quite lengthy and sometimes the motivation is not so clear. For example, Section 3.1, 3.2, 3.3 only provides background, and then Section 3.4 (which appears quite late in the text) actually tackles the problem.

- The experiments are only reported with quantitative results. Having some visualizations of the segmentation and depth and some novel view renderings in the results would be nice.

- When comparing segmentation results, please add IoU metric.


Rating

The paper sounds weak in the sense that some of its contributions are on improving NeRFs while the decomposition part is rather marginal. The presentation is sometimes misleading. There are a few concerns about the experiments and baselines as well. Therefore, my current rating is below the acceptance threshold. I invite the authors to try considering the questions as commented.

---

> ### Author Response · Authors · 2021-11-11
> **Initial Response**
>
> Thank you for your thoughtful review. While we prepare a revised version of the paper, please find our initial responses below.
>
>  - Could you please clarify the following statement: “However, it lacks the motivation of leveraging nerf lack. The authors must make it clear.”
>  - **Clarification of the goal**: We would like to clarify that our goal is to learn to jointly infer (in the sense of 3D reconstruction) and segment the 3D geometry of a scene, given a 2D image (without depths). The trained model can infer separate scene functions for different parts of the scene. These scene functions provide a segmentation of the whole volume (since we can query each scene function separately to produce density & colour values at any 3D location). We can still render 2D segmentations from arbitrary viewpoints as shown in Fig.1 and section 3.4, but this is an auxiliary result.
>  - **Relation to Depth-supervised NeRF**: We disagree that our formulation is similar to that of Deng et al. They still do ray marching, requiring >100 scene function evaluations per ray, and simply place a MSE loss on depths. In contrast, our depth loss eliminates the need for ray marching, requiring only two scene function evaluations per ray. It will therefore be much faster when dense depth supervision is available. We will add an ablation of our model, where we use the formulation from Deng et al. Regarding novelty, we would also like to note that Deng et al. only appeared recently as a preprint, did so after the preprint for this work, and also after the cutoff date set by the [ICLR reviewer guidelines](https://iclr.cc/Conferences/2022/ReviewerGuide) for work considered contemporaneous. We will therefore cite it as concurrent work.
>  - **Depth Training & Unsupervised Segmentation**: ObSuRF can segment scenes without using depth for training, though this results in a significantly increased computational cost of model training. We will provide an ablation in the updated version of the paper.
>  - **Unbiased Estimator**: The fact that the proposed objective $\log p(C, t) = \log p(C | r(t), d) + \log \sigma(r(t)) + \log T(t)$ is unbiased follows immediately from the observation that the first two summands can be evaluated exactly, and the last one only consists of a single integral, which may be estimated using Monte Carlo sampling (i.e., there is no nested integration, which is what biases the original objective).
>  - **Qualitative results**: Please watch the videos on the anonymous project site linked in the paper: https://sites.google.com/view/obsurf/ They illustrate reconstructions, depth estimates, and segmentations from arbitrary viewpoints.
>  - **Depth Estimation**: We do not claim to improve SOTA on depth estimation as depth is simply an emergent quality of NeRF rendering; we provide errors of the estimated depth just as additional information.
>  - **Comparison with other segmentation methods**: We compare with SOTA models in unsupervised 2D segmentation. Unsupervised volumetric 3D segmentation based on 2D input is a new setting for which baselines were not available at the time of writing.
>  - **Real-world Results**: Scaling to real world data is an ongoing effort in the unsupervised segmentation community. Only very recent papers have included results on simple real world data, such as GENESIS-V2 (Engelcke et al., 2021). Similarly, only very recent and limited results exist for amortized multi-scene NeRFs inferring NeRF scene functions for real world data: both pixelNeRF (Yu et al.) and GRF (Trevithick and Yang) produce blurry single-object ShapeNet reconstructions. pixelNeRF provides real-world results, but these are also blurry, especially in the case where only one input image is provided. Another issue is the lack of 3D multi-object datasets with a sufficient (>1000) number of scenes. Google Scanned Objects contains only 1033 objects which by themselves would not allow segmentation and lead to overfitting. Creating a multiobject dataset based on GSO is a good idea for future work.

---

> ### Author Response · Authors · 2021-11-29
> **Revision**
>
> Dear Reviewer VrrK,
>
> Thank you again for your review of our work. In our revised version of the manuscript, we have incorporated your feedback by adding both standard NeRF training, as well as depth-supervised NeRF training following Deng et al. as baselines, including a comparison of training times. We have also added ablations of our model which lack pixel conditioning, overlap loss, or both. We also now provide detailed qualitative results for each model type on the [project website](https://sites.google.com/view/obsurf/).
>
> As the discussion period is approaching its end, we would be grateful if you could confirm whether our responses and the additions to the manuscript have addressed your concerns, and let us know if any issues remain.

---

### Official Review · Reviewer_1keq · 2021-11-02

**Correctness:** 3
**Technical Novelty And Significance:** 4
**Empirical Novelty And Significance:** 4
**Recommendation:** 8
**Confidence:** 2

**Details Of Ethics Concerns:**

none that I can see

**Main Review:**

I like the idea to use more information allows to use a more efficient algorithm. The topic is out of my field yet the derivations seem theoretically sound. The experimental evaluations (using simulated data) demonstrate that the algorithm works, at least on very simple data.

I have a few questions and remarks:

- What does "Eq (3) is renormalized" mean? Please explain that procedure.

- "a significant chance that thin, high-density volumes are missed": Standard hardware for capturing RGB-D data likely does not capture thin, high-density volumes either, thus the proposed approach would not be successful for such data. This somewhat implicit argument (the proposed approach can solve thin, high-density volumes) should be removed, thus.

- Can the authors drop a few words about the derivation of log p?

- Eq (4): Why are 50% of the samples taken from the last 2% of the ray? What happens is the ray is sampled differently?

- maximizing Eq 4 under a Gaussian distribution: Is this assumption justified? Can it be violated? If so, which attributes would such a part of a volume have?

- How is the joint log-likelihood p(t,C) defined? Why are only 2 evaluations of the NeRF necessary?

- What do the authors mean by "we use the code to shift and scale the activations at each hidden layer"?

- By Fig 1, I assume that the number of objects is provided for each image? Stated a bit differently, it seems that differently many slots are used for the input images? That reduces the claim in the title about the unsupervised segmentation since providing the number of objects can be considered as a weak supervision signal.



**Summary Of The Paper:**

This paper is about unsupervised segmentation both of 3D volumes and of images. The idea is to learn a set of Neural Rendering Fields (NeRFs) which explain a given image. To reduce computationally effort, the authors assume RGB-D data and show how that reduces the computational burden by two magnitudes. The proposed algorithm can learn segmentation masks and a set of NeRFs that generate each individual object and the background.

**Summary Of The Review:**

A major disadvantage of using NeRFs is their need for very expensive GPUs. I like that the authors address this issue and propose a solution. The price that one has to pay for that (RGBD instead of RGB, providing the number of objects) can be justified, often. Furthermore, I suppose that this paper will trigger follow-up works which improve the weaknesses of this work.

---

> ### Author Response · Authors · 2021-11-11
> **Initial Response**
>
> Thank you for your thoughtful review. While we prepare a revised version of the paper, please find our initial responses below.
>
>  - **Renormalization**: By renormalization we mean that estimates for $p(t)$ will be divided by the transmittance of the ray in the sampled interval $T(t_{\text{far}}) = p(t < t_{\text{far}})$, such that they add to one. We will clarify in the revision.
>  - **Thin volumes**: By “thin, high-density” volumes, we are referring to things like a wooden tabletop: When the vertical distance between samples during NeRF rendering is more than a few centimeters, there is a significant chance that none will fall into the tabletop volume, and it will not be captured by rendering. But, since it is not translucent, it will be easily captured by RGBD hardware.
>  - **Importance Sampling**: The importance sampling scheme is designed to lower the variance of the gradient estimate. The model will learn very early in training that the majority of the space between the camera and the ground plane is always empty. It is therefore more fruitful to collect samples near object surfaces, where most errors will occur. Without importance sampling, training will still work, but it will be slower due to higher variance in the $\log p(t)$ estimates.
>  - **Joint Likelihood**: By the chain rule, the joint probability $p(t, C)$ is simply the product of $p(t)$ (Eq. 2) and $p(C | r(t), d)$ (section 3.3). One sample is required for each of the two parts of Eq. 4: $\sigma(r(t))$ is the density at the surface, and the expectation is estimated with importance sampling, by using one sample between surface and camera. Taking additional samples is not necessary (though it would result in a lower-variance estimate) since this yields an unbiased estimate of $\log p(t, C)$.
>  - **Conditioning on $z_i$**: The activations are adjusted as follows: $h’ = (h + \beta) * \alpha$, where $\alpha$ and $\beta$ are vectors predicted from $z_i$ using a linear map. This is detailed in appendix B.2.
>  - **Number of Slots**: The only information provided is an upper bound for the number of objects per scene in a given dataset, which is used to set the number of slots. Allocating those slots to objects, leaving some of them empty if necessary, is an emergent quality of the model.

---

### Official Review · Reviewer_bAmB · 2021-11-03

**Correctness:** 3
**Technical Novelty And Significance:** 2
**Empirical Novelty And Significance:** 3
**Recommendation:** 6
**Confidence:** 3

**Main Review:**

I found the paper a bit hard to follow, as it heavily relies on previous works. However, I think successfully builds on previous works and shows promising results on the used synthetic data.

Strengths:
* Using depth to reduce the number of samples considerably is an intuitive way to reduce the computational complexity of training NeRFs.
* The proposed method considerably outperforms (quantitatively) the previous methods under comparison in Tables 1 and 2.

Weaknesses:
* The segmentation part seems to be already proposed in  "Object-centric learning with slot attention (Neurips2020)". What is the difference?
* The conclusion section could be improved to be more specific on the main paper achievements/observations.
* Visual comparison against the previous methods in Table 1 and/or 2 are missing.
* No results on real-world data are provided.
* Similarly, the results seem to be only good on the CLEVR dataset which contains very simple geometries. For the ShapeNet dataset, the results seem to be rather blurred.

**Summary Of The Paper:**

The authors presented a method to infer a neural radiance field per object in a scene from a single input view.
uses  Object-centric learning with slot attention

The proposed architecture takes the shape of an autoencoder. The encoder side takes a single image and pose as inputs and generates N latent codes for each object in the scene following  Object-centric learning with slot attention (NeurIPS2020). The latent codes condition shared NeRF decoders to define the geometry and appearance of each object. The full scene is reconstructed by superposing the multiple Radiance Fields.

Furthermore, to reduce the computational complexity of NeRFs, the authors propose to use GT depth maps to avoid full ray marching.

**Summary Of The Review:**

The results shown in this work seem promising and advance towards single view 3D scene understanding. However, due to the mentioned weaknesses in the main review, I am leaning towards a "marginally below the acceptance threshold" score.

Update:

The authors have addressed the comments in my initial review and showed improved results on the Shapenet dataset. Therefore I am updating my recommendation.

---

> ### Author Response · Authors · 2021-11-11
> **Initial Response**
>
> Thank you for your thoughtful review. While we prepare a revised version of the paper, please find our initial responses below.
>
> - **Slot Attention**: While we use a version of slot-attention from Locatello et al. for unsupervised segmentation, the original work applied slot-attention only to 2D images. The heart of our contribution is to demonstrate how to jointly infer and segment the 3D geometry of a scene given a single image and, in particular, how to train such a model efficiently.
>  - **Visual comparison**: We will add visualisations of ObSuRF’s 2D results to the appendix. For the baseline models, we will have to refer the readers to the respective papers, since we do not have access to their implementations.
>  - **Real-world Data & Blurry Results**: Scaling to real world data is an ongoing effort in the unsupervised segmentation community. Only very recent papers have included results on simple real world data, such as GENESIS-V2 (Engelcke et al., 2021). Similarly, only very recent and limited results exist for amortized multi-scene NeRFs  for real world data: both pixelNeRF (Yu et al.) and GRF (Trevithick and Yang) produce blurry single-object ShapeNet reconstructions. pixelNeRF provides real-world results, but these are also blurry, especially in the case where only one input image is provided. Another issue is the lack of 3D multi-object datasets with a sufficient (>1000) number of scenes.
>  - We will update the paper with less blurred results obtained via better use of pixel conditioning.

---

> ### Author Response · Authors · 2021-11-29
> **Revision**
>
> Dear reviewer bAmB,
>
> Thank you again for your review of our work. In our revised version of the manuscript, we have incorporated your feedback by adding visual results on the 2D datasets and clarifying the relationship to slot attention. We also provide new qualitative results on the [project website](https://sites.google.com/view/obsurf/) illustrating how blurriness on Shapenet is reduced via pixel conditioning.
>
>  As the discussion period is approaching its end, we would be grateful if you could confirm whether our responses and the additions we have made to the manuscript addressed your concerns, and let us know if any issues remain.

---

### Official Review · Reviewer_8UAh · 2021-11-03

**Correctness:** 4
**Technical Novelty And Significance:** 3
**Empirical Novelty And Significance:** 3
**Recommendation:** 6
**Confidence:** 3

**Main Review:**

The paper is well written, is pleasant to read and the formulation seems sound. The authors clearly justify their decisions and overall their work provide valuable insight about the problems of competing approaches in the literature. The experimental validation seems well designed and the results out-perform the state of the art for both 2D and 3D datasets.

Regarding the novelty of the work, there are two main aspects: First, the unsupervised decomposition of the scene, which is somehow incremental, given that it's achieved by applying the slot-based approach of [Locatello et al. 2020] to NeRFs, and also very related to [Yu et al. 2021]. Second, the training speed-up using depth training data, which seems to be the most significant contribution. However, this was also explored in a recent work by Kangle Deng et al. "Depth-supervised NeRF: Fewer Views and Faster Training for Free".

I do understand that since this work was published in arxiv in July 2021, according to ICLR guidelines the authors are not required to compare their work against it but given that the most significant contribution is so similar, I would find necessary to reference the paper and briefly mention it in the related work.

The authors comment on the fact that it's possible that some of the reconstruction error gains are due to the additional (depth) supervision. This renders the comparison a bit unfair given that the present work is using additional training data. I miss further discussion, in a similar fashion as the ablation experiment with the overlap loss, but focusing on the depth training data.


**Summary Of The Paper:**

The authors propose a method to synthesize novel views of 3D scenes while inferring a decomposition of the scene into multiple objects.  The compositional reasoning is achieved by a slot-based encoder so that there's no need for specific supervision on the object categories. The authors also contribute with a novel loss function that exploits depth training data to improve the sampling strategy and therefore increasing training speed as well as lowering the reconstruction error. The results out-perform the state-of-the art, and are validated against 2D and 3D datasets.

**Summary Of The Review:**

I would recommend this paper for acceptance as long as no other reviewers have major concerns. The paper conveys a good contribution. It's quite unfortunate that those two recent works (Kangle Deng and Yu et al. , both in July 2021) clash significantly with the two main novelties of this work, but as stated above, those papers are too close to the submission deadline to be considered double submission. In any case, I find the citation of the work of Kangle Deng. et al. a requirement for acceptance.

---

> ### Author Response · Authors · 2021-11-11
> **Initial Response**
>
> Thank you for your thoughtful review. While we prepare a revised version of the paper, please find our initial responses below.
>
>  - **Training NeRF with Depth**: We will add the reference to Deng et al., but would like to point out that their formulation has very different performance characteristics from ours. They still do ray marching, requiring >100 scene function evaluations per ray, and simply place a MSE loss on depths. In contrast, our depth loss eliminates the need for ray marching, requiring only two scene function evaluations per ray. It is therefore much faster when dense depth supervision is available. We will add an ablation of our model, where we use the formulation from Deng et al.
> - **Slot Attention**: While we use a version of slot-attention from Locatello et al. for unsupervised segmentation, the original work applied slot-attention only to 2D images. The heart of our contribution is to demonstrate how to jointly infer and segment the 3D geometry of a scene given a single image and, in particular, how to train such a model efficiently.
>  - **Amortized NeRF Training**: Yu et al. present a method for amortizing NeRF training, yielding an image conditioned model, but they do not attempt to segment or understand the inferred scene representation.
>  - In summary, our work puts together a number of contributions from existing literature (amortized NeRF training on many scenes, unsupervised segmentation, single-image 3D reconstruction) into a single model. In addition, we show how to do unsupervised 3D segmentation using NeRFs. This is absent from the current literature: there are GANs which parametrize objects by separate NeRFs but they do not segment existing scenes. We also show how to train such a model efficiently (also absent).

---

> ### Author Response · Authors · 2021-11-29
> **Revision**
>
> Dear Reviewer 8UAh,
>
> Thank you again for your review of our work. In our revised version of the manuscript, we have incorporated your feedback by adding both standard NeRF training, as well as depth-supervised NeRF training following Deng et al. as baselines, including a comparison of training times. This quantifies the impact of incorporating depth supervision, as well as the relative performance of our depth loss compared to that of Deng et al. We also provide detailed qualitative results for each model type on the [project website](https://sites.google.com/view/obsurf/).
>
> As the discussion period is approaching its end, we would be grateful if you could confirm whether our responses and the additions to the manuscript have addressed your concerns, and let us know if any issues remain.

---

### Author Response · Authors · 2021-11-29
**Revision**

Dear reviewers,

we have updated the manuscript based on your feedback. These are the main changes:

 - We have added baselines to Table 2 in which ObSuRF is trained using standard NeRF training and depth supervised NeRF training (Deng et al.). While they achieve comparable results on CLEVR3D, they fail to produce accurate geometry or segmentations on MultiShapenet. We also provide learning curves in the appendix (Figure 4), which support the argument that our depth loss formulation allows for much faster and effective training on the difficult MultiShapenet dataset.
- In Table 2, we also provide ablation results for ObSuRF trained without pixel conditioning (Yu et al.), without the overlap loss $\mathcal{L}_O$, or both.
- The project website (https://sites.google.com/view/obsurf/) now provides videos rendered from all six models, accessible through the menu at the top. In particular, note the more accurate geometry produced by ObSuRF when pixel conditioning is used, at the cost of slightly worse color segmentation.
- We have added visualizations of ObSuRF's output on the 2D datasets to the Appendix (Figure 3).
- We have updated the text to reflect these changes and your feedback, in particular discussing the relation to Deng et al., and clarifying previously unclear technical details.

Thank you again for the useful feedback. We hope these changes address your concerns.

---

### Decision · Program_Chairs · 2022-01-20

**Decision:**

Reject

**Comment:**

The paper develops a method for decomposing 3D scenes into objects by coupling NeRF decoders to representations produced by a slot-based encoder.  After the discussion phase, reviewer ratings are mixed with three on either side of above/below threshold, and one higher (but low confidence) accept score.

Drawbacks include limited novelty, as stated by Reviewer 8UAh: "the unsupervised decomposition of the scene, which is somehow incremental, given that it's achieved by applying the slot-based approach of [Locatello et al. 2020] to NeRFs".  Reviewer bAmB likewise mentions this issue.  Reviewer VrrK: "some of its contributions are on improving NeRFs while the decomposition part is rather marginal". Reviewer VrrK also raises concerns about lack of experiments on real data: "Since the proposed method is based on NeRF, how well does it work with real photographs?"

The AC agrees with the marginal rating of the reviewers and is particularly concerned with overall novelty of the proposed pipeline and question of applicability beyond simulated data.  More work seems required to solidify an experimental case on real images.